# Ion Substitution-Induced Distorted MOF Lattice with Deviated Energy and Dielectric Properties for Quasi-Solid-State Ion Conductor

**DOI:** 10.3390/nano15040274

**Published:** 2025-02-11

**Authors:** Yike Huang, Yun Zheng, Yan Guo, Qi Zhang, Yingying Shen, Hebin Zhang, Yinan Liu, Yihao Zheng, Pingshan Jia, Rong Chen, Lifen Long, Zhiyuan Zhang, Congcong Zhang, Yuanhang Hou, Kunye Yan, Ziyu Huang, Manting Zhang, Jiangmin Jiang, Shengyang Dong, Wen Lei, Huaiyu Shao

**Affiliations:** 1Joint Key Laboratory of the Ministry of Education, Institute of Applied Physics and Materials Engineering, University of Macau, Taipa, Macau SAR 999078, China; 2Jiangsu Province Engineering Laboratory of High Efficient Energy Storage Technology and Equipments, School of Materials Science and Physics, China University of Mining and Technology, Xuzhou 221116, China; 3Jiangsu Key Laboratory of New Energy Devices & Interface Science, School of Chemistry and Materials Science, Nanjing University of Information Science and Technology, Nanjing 210044, China; 4The State Key Laboratory of Refractories and Metallurgy, Wuhan University of Science and Technology, Wuhan 430081, China

**Keywords:** solid state, electrolyte, ion exchange, battery, dielectric

## Abstract

Solid-state electrolytes are currently receiving increasing interest due to their high mechanical strength and chemical stability for safe battery construction. However, their poor ion conduction and unclear conduction mechanism need further improvement and exploration. This study focuses on a hybrid solid-state electrolyte containing MOF-based scaffolds, using metal salts as the conductor. In this paper, we employ an ion substitution strategy to manipulate the scaffold structure at the lattice level by replacing hydrogen with larger alkali cations. The research systematically presents how changes in the lattice affect the physical and chemical properties of MOFs and emphasizes the role of scaffold–salt interactions in the evolution of ion conduction. The results reveal that long range-ordered structural distortion can enhance permittivity at 1 Hz, from 58 ohms to more than 10 M ohms, which can boost ion pairs dissociation and improve the transference number from 4.7% to 22.6%. Defects in the lattice can help stabilize the intermediate state in the charge transfer process and lower the corresponding impedance from 2.6 MΩ to 559 Ω.

## 1. Introduction

Batteries for mobile applications are now a fundamental part of our daily lives. However, their high energy capacity brings up safety issues, and is becoming increasingly concerning [1]. Employing solid-state electrolytes and constructing solid-state batteries are both major topics in the battery industry [2,3,4,5]. Using solid-state electrolyte batteries with high mechanical strength and a less flammable nature can improve the safety of mobile devices. Furthermore, their unique electrolyte–anode interfaces can also suppress the growth of metal dendrites as seen in metal anode-based batteries, and reduce the risk of short circuits from dendrite penetration [6,7,8,9]. However, compared with liquid electrolytes, solid-state electrolytes suffer challenges in their ionic conductivity and interface resistance. In addition, their ability to dissociate ion pairs into sole metal cations for electrochemical reactions is also critical for their use [10,11].

Generally, there are two strategies for constructing solid-state electrolytes. One is employing ceramic or ionic compound-based solids which have conductive cations in their lattice (and therefore do not need additional carrier cations), such as NASICON [12], sulfide-based compounds [13], metal halides [14], and garnet-like oxides [15]. The other is using solid scaffolds with metal salts which serve as conductive ions. Scaffold materials may be polymers such as PEO [16], gel polymer composites [17], or MOFs [18,19,20,21]. For scaffold–salt electrolytes, the interaction between the scaffold and salts decides the bulk conductivity and the transference number, which is an indicator of the utilization ratio of specific cations for the electrochemical reaction. Currently, a full understanding of the ion conduction mechanism of scaffold–salt electrolytes is still far from a conclusion, due to the difficulty in characterization and the complexity of scaffold structures, especially those of polymers [22]. To this end, researchers have made continuous efforts to explore various aspects, such as scaffold modification, changing scaffold–salt interactions, and the final electrochemical performance [23,24]: Huang et al. designed a ferroelectric polymer and found that its high dielectric properties can benefit both conduction and its transference number [25]; Wang et al. modified cellulose by esterification, and concluded that the breakdown of the raw hydrogen bond network and coordination effects between oxygen sites and cations contributed to its conductivity [26]; Xu et al. designed a polymer with a coaxial MXene arrangement and found that polymer–MXene interactions can affect the F-H and -CN functional bonds, causing a fascinating Li conduction effect [27]. However, the current research still lacks a systematic study on how each property affects the scaffold–salt interaction and the final electrochemical results. Most studies have focused on one specific modification, which results in a lack of data on a gradual evolution of the performance. Therefore, we aim to bridge the relationship between the micro structure and the ionic conduction behavior, which may benefit the mechanical study of solid-state electrolytes and device designations for the energy storage industry.

In this work, we choose the MOF MIL-121 as the scaffold to study the relationship between the lattice, spectrum properties, dielectric performance, and the electrochemical performance. MIL-121 was modified using a simple and moderate ion substitution process generating six kinds of MOF analog, as presented in Figure 1. Depending on the individual chemical and physical properties of the inserted ions, the MOF lattice was distorted to different extents. These distortions led to a difference in other electrochemical and physical properties. We systematically studied how those modifications affected the MOFs’ structure, and how those differences in the scaffold structure brought out different ion conduction properties.

## 2. Experimental

### 2.1. Synthesis

MIL-121 was prepared using a hydrothermal approach. Briefly, 5.328 g Al(NO_3_)_3_·9H_2_O was dissolved in 22.2 mL H_2_O with 1.776 g 1,2,4,5-benzenetetracarboxylic acid (H_4_BTeC). The mixture was heated at 210 °C for 24 h following filtration and dried at 90 °C for 12 h. The resulting white powder was MIL-121 (AlBTeC). To modify the MIL-121, 0.5 g MIL-121 was dispersed into 25 mL buffer containing 0.0227 mol 3-morpholinopropanesulfoinc acid (MOPS) and 0.0204 mol base to reach a pH of 7.5–7.8. For each cation substitution, the bases we used were LiOH·H_2_O, NaOH, KOH, Rb_2_CO_3_ (0.0102 mol), and CsOH·H_2_O. The MIL-121 powder was suspended in the buffer and kept stirring for 16 h at room temperature. The concentration for the substitution solution was MIL-121 20 g L^−1^; MOPs 0.908 mol L^−1^, and alkali base 0.816 mol L^−1^ (for Rb_2_CO_3_, 0.408 mol L^−1^). After that, we filtrated, washed, and calcinated the products at 300 °C for 300 min with a heating rate of 5 °C min^−1^.

To mix the modified MIL-121 with electrolytes, 200 mg of modified MIL-121 was mixed with 0–200 mg magnesium bis(trifluoromethanesulfonimide) (MgTFSI) along with 0–25 μL triethylene glycol monomethyl ether (T3GM) through 4 h ball-milling. The milling speed was 500 rpm min-1 with one Φ13 mm and three Φ6 mm ZrO_2_ balls. The milled powder was pressed to Φ10 mm, 0.9~1.1 mm thickness plate for electrochemical tests. A simple schematic diagram of the preparation (the case of Li-substitution) is in Figure 2.

### 2.2. Electrochemical Measurements

The EIS spectra were obtained via Solartron 1260A (AMETEK, Berwyn, PA, USA), with a frequency from 10 MHz to 1 Hz. The electrolyte plates were clamped using 2 stainless steel plates as the electrode and were assembled to a CR2032 coin battery for testing. All of the cells were constructed under argon atmosphere with oxygen and H_2_O contents lower than 0.01 ppm, and all the measurements are conducted at room temperature.

The potentiostatic method for transference number measurement was realized by EIS and potentiostatic tests by CHI760E. Electrolytes were also assembled into a CR2032 battery, but the stainless steel electrode was changed to a polished Mg plate. For the potentiostatic tests, the discharge voltage was set at 0.30 V for 1 h. The transference numbers (*t*) were calculated with the following formula:$$t=\frac{I_{s}R_{b}^{0}[\Delta V-I_{0}R_{i}^{0}]}{I_{0}R_{b}^{s}[\Delta V-I_{s}R_{i}^{s}]}$$

In which Δ*V* is the potentiostatic voltage of 0.30 V, and *I*_0_ and *I*_s_ are the current at the beginning and end of the potentiostatic test. *R*_b_ is the bulk ohmic resistance and *R*_i_ is the interface resistance, with mark 0 and s representing the beginning and end of the potentiostatic test, respectively.

## 3. Results and Discussion

Briefly, the raw AlBTeC MOF was obtained through a hydrothermal approach. The ion substitution was conducted by immersing the MOFs into an ion-rich solution with pH at 7.5–7.8, which was controlled by the buffer. The element contents were further confirmed by ICP-MS, and the data can be seen in Appendix A. The raw data were calculated as weight ratios. The aluminum contents for the raw AlBTeC MOFs were about 8.9 wt.%, which did not change a lot after substitution. To give a comparison of how many ions for each alkali element were substituted into the MOFs, we calculated the data in mole ratios, and divided the alkali element contents by their aluminum contents, as presented in Table 1. One can see that the substitution contents decreased as the alkali elements’ periods increased. This implies that the larger cation radius hindered the ions inserted into the MOFs. We deduced that the larger ion blocked the pathway for further insertion, or possibly that the large structural distortion brought extra repulsion, hindering higher substitution. All of the alkali contents were below 2.0, which is below the theoretical limit of one aluminum with two carboxyl hydrogen from the chemical formula Al(OH)[H_2_BTeC] [28]. This is in agreement with the hypothesis that doped alkali cations replace the carboxyl hydrogen from the [H_2_BTeC] legends.

The XRD data are presented in Figure 1a and Appendix A. In Appendix A, the as-prepared MOF showed a diffraction well matched with that which was simulated using the structural information in [28]. In Figure 1a we can see that most of the diffraction features remained after substitution. This means that ion exchange did not change the basic crystal symmetrical structures. The substituted alkali ions change the peak position and intensity of the XRD curves. For the peak position, we collected the 2*θ* data of the strong and distinguishable peaks in Appendix A. Based on these data and the Debye–Scherrer equation, the interplanar spacing could be calculated. To clearly understand the interplanar distortion after substitution, the spacing *d* was ratioed by the *d*_raw_ of the non-modified MOF and presented in Figure 1b. This shows that some interplanar structures increased their spacing but some decreased, implying a distortion in the lattice. All of the lattice distortion is less than 3%, implying that the robust MOF lattice did not collapse. We expected the lattice distortion to follow the trend of ion radius, which corresponds to a continuously increasing or decreasing curve for each interplanar structure. However, the results surprisingly fluctuated: for example, in the (100) interplanar spacing, the spacing ratio decreased, increased, decreased again, slightly increased, and decreased. The data indicated that the impact of the doped ion radius on the lattice was not linear. For Cs, its large size brought the most distortion to the lattice, but the smallest Li^+^ ion also showed a considerable distortion ratio, which may be due to its high doping amounts, as confirmed by ICP-MS. Figure 1b reveals that the order of distortion extent was Cs > Li ≈ K ≈ Rb > Na. Besides the peak position, the XRD peak intensity can also reveal structural information; we present the diffraction intensity data in Figure 1c. The XRD peak intensity can indicate the crystallization degree of the composites. Generally, the chaos in an interplanar structure, including fluctuated spacing and deviated atom positions, may decrease the XRD peak intensity. In another words, the crystal phase becomes amorphous. All of the modified MOFs showed inferior peak intensity compared with the original structure. Figure 1c showed that the Li and Cs-modified samples displayed the lowest intensity, implying the lowest regularity in lattice due to the non-uniform distortion attributed to high Li loading and large Cs volume. The K-modified sample maintained the most diffraction intensity, implying a relatively regular lattice. We deduced that there were two possible major factors for spacing distortion: one is the volume effect, i.e., the larger volume of the alkali ion leading to an expansion; the other is the bonding effect, i.e., the ion substitution disrupted the ligand electron distribution through the carboxyl groups, and affected the conjugated structure, the π-stacking structures, and the lattice spacing. The distorted lattice may change the scaffold–electrolyte interactions, such as exposed defects and ionized surface for attracting ions, and lead to a new mechanism or a different electrochemical response of the final hybrid electrolyte.

Before we employed the MOFs as an electrolyte scaffold, the MOFs were dehydrated to prevent water residues from corrupting the Mg electrode into inert MgO. We heated the MOFs at 300 °C before further treatment. The XRD results of dehydrated MOFs are presented in Figure 1d to verify the changes in crystal structure after heating. Compared with the as-prepared MOFs, there are no new peaks observed. This implies that the 300 °C treatment did not generate new phases such as alkali oxide and aluminum oxide, which means that the substitution and the heating treatment did not decompose the MOFs. The diffraction intensity profiles are shown in Figure 1e. For non-modified MOFs, the diffraction intensity increased after heating, which follows the general rule that calcination benefits crystallization; during calcination, the atoms moved to a thermodynamically stable position. For the raw MOFs, the raw MOF lattice was the thermodynamic stable structure. Thus, such moving could repair the atom deviation caused by distortion and re-build the long-range crystallographic order, resulting in increased XRD peak intensity. Figure 1e showed that the peak intensity of the Li-modified MOF increased, but that of the others decreased. For the Li-modified sample, the raw MOF lattice was the thermodynamic stable structure. Benefiting from the small size of the Li ion, the raw lattice was still the most stable structure, without too much repulsion from the substituted ions. Although the Li substitution (with the help of the largest amount of substitution) brought considerable lattice distortion, such a distorted lattice was not stable and could be alleviated by calcination. For the other ion-substituted MOFs, the calcinated samples showed decreased intensity, implying a worse crystallographic order. During calcination, atom vibration can lead the atoms to move to their most thermally stable positions. In the case of raw MOF and Li-modified MOF, the thermally stable positions followed the raw MOF lattice positions. Thus, the calcination re-built the crystallographic order. However, for the other, larger alkali ions, the decreased crystallographic order indicated that the raw MOF lattice positions were not the most thermally stable. This may be because the larger alkali ion may generate great repulsion if in its raw lattice positions, and a further distorted position is more thermally stable. To give a simple summary, all of the alkali ion-modified MOFs showed a lattice distortion. The extent of distortion depended not only on the ion size but also the doping amount. For example, the smallest Li ion, with the help of large doping amount. showed a stronger distortion than the Na ion. The interplanar spacing distortion was in the order of Cs > Li ≈ K ≈ Rb > Na. After calcination, the distortion was alleviated for the Li-modified sample, but the others showed stronger distortion.

The XRD results present the lattice-level structural evolution of cation exchange and the following calcination. To further understand the structural changes on a particle-level, we employed SEM to obtain morphological information. As presented in Figure 2a, raw MIL-121 has a short stick morphology with sharp edges. The alkali ion substitution did not change the macro-morphology from Li to Cs. Although the XRD indicated that the insertion of Cs decreased the crystallization significantly, the SEM proved that the structure, on a particle level, remained almost unchanged. After heating, the edges and the stick shape remained sharp and clear without distortion, implying a good heat resistance at 300 °C (Figure 2b). Even for the Cs ion, the particles are still in the stick shape with sharp edges, despite the significant attenuation in X-ray diffraction. These findings indicate that the alkali ion substitution changed the local symmetry on the lattice scale but did not affect the macro-morphology, and that the particle structures showed a good stability in substitution and calcination.

We also employed the EDS measurement to confirm the elemental distribution. As presented in Figure 2c, we can see the element distribution of Al, O, and alkali ions following the MIL-121 particle’s stick shape in the electron image. In addition, the signals for S and N, which were contained in the counter-anion MOPS (the agent for maintaining pH in the ion exchange), are below the background limit. The results support that only the alkali cation was doped into the MOFs, with a homogeneous distribution. The buffer anion in the ion exchange reagent did not remain in the MOFs.

To further confirm the chemical bonding features of the introduced alkali cations, we employed FT-IR to describe the bonding evolution after doping [29]. As presented in Figure 3a, we can see that all of the cation-substituted samples maintained almost all of the IR features from the raw AlBTeC MOF. This further verifies the conclusion that the introduction of alkali cations did not decompose the critical MOF skeleton but disordered its lattice. Moreover, we can observe peaks split at 1350 and 1405 cm^−1^, corresponding to δ COH and ν_s_ OCO. One can clearly observe these peaks in Figure 3b, which shows an enlarged FT-IR plot; in contrast, the raw MOF presented a smooth single peak at these positions. Both peaks are carboxyl group-related, and this provides strong evidence that it represents doped alkali cation-substituted carboxyl hydrogen. As revealed by ICP-MS, the mole ratio of aluminum ions and doped cations is 1:1.072 to 1:0.625. For the raw MOFs, the mole ratio of aluminum ions and non-coordinated carboxyl hydrogen is 1:2, according to the molecular formula Al(OH)[H_2_BTeC] [28]. If the doping took place on the carboxyl hydrogen, there would be raw carboxyl groups and alkali-cation-doped carboxyl groups co-existing in the doped MOFs, which may be attributed to the peak splitting on the FT-IR spectrum. Such peak splitting was confirmed, and the finding was further strengthened by the solid-state ^13^C NMR result in Figure 3c. For the raw AlBTeC MOF, there are two major chemical shift features in the spectrum: one involves the peaks at 160 to 180 ppm, corresponding to the carbon from the benzene structure; the other can be seen at 120 to 150 ppm, corresponding to the carboxyl groups. For the AlBTeC MOF, the ligand H_4_BTeC, i.e., 1,2,4,5-benzenetetracarboxylic acid, has only two kinds of carbon chemical environments that can be distinguished by NMR: the benzene carbon and the carboxyl carbon. After alkali cation insertion, the peaks at 160 to 180 ppm (benzene carbon) did not split, whereas significant splitting was observed in peaks at 120 to 150 ppm (carboxyl carbon). This reveals that after modification, a part of the carboxyl carbon shifted to a new chemical environment. The co-existence of two types of carboxyl carbon contributed to the split NMR peaks. Based on these FT-IR and NMR results, we can conclude that the doped alkali cations mainly occupied the non-coordinated carboxyl hydrogen position in Al(OH)[H_2_BTeC].

Besides the lattice, macro-structure, and chemical bond structure, we observed a color change of the substituted MOFs, as presented in Figure 4a. The raw MOF, AlBTeC, showed a white color in its powder. This is a typical color of the +3 valanced aluminum composite. After cation substitution, the MOFs became yellow. The Li-doped sample showed a light-yellow color, and Na became more saturated. It seemed that the K-doped MOF showed the brightest yellow, but saturation differences among Na, K, Rb, and Cs were hard to distinguish by optical observation. Such a color change can be attributed to the orbital splitting caused by the substitution of alkali cations; thus, a study on the color could give us information about the interaction between alkali cations and the MOFs. To quantitively describe the color change, we employed UV–Vis spectroscopy. The MOF samples were adhered on a tape for testing, employing a spectroscopy device equipped with an integration sphere. For the raw AlBTeC MOFs, the largest decrease in reflectance occurred below 300 nm, which verified its optical white color. This peak could be attributed to the π-π* transition of the benzene and carboxyl groups. For the ion-doped samples, all of the samples showed a red shift, with a shifted peak position at 350 to 400 nm. We deduced two possible reasons for this shift: one is that the alkali elements which substituted hydrogen performed stronger electron donating properties that benefited the stabilization of the excited π* state and caused the red shift; the other is that the lattice distortion, as confirmed by XRD, brought a similar stabilization effect and the red shift. As marked in Figure 4c, the right shift of the absorption peaks was in an order of K ≈ Rb > Na > Cs > Li, implying that the K substitution lead to the most stable excited π* state. Furthermore, the K-modified sample also performed the highest peak intensity, indicating that the energy-state influence widely existed over the MOF structures.

Before the MOFs were employed as electrolyte scaffolds, their original electric properties were considered. Especially, their dielectric properties and permittivity were proved to have profound influence on ion conductance. Huang et al. reported that high permittivity of the polymer benefits ion dissociation and the conduction of the final polymer-based electrolyte [25,30]. They modified the structure and organic groups of the polymer to improve its dielectric properties. There have also been reports that adding high dielectric ceramic contents into polymers can improve its ion dissociation properties [31]. The permittivity of our MOFs was measured using electrochemical impedance spectroscopy (EIS). As presented in Figure 4d, the raw MOF showed a permittivity lower than 100 ohms over the whole frequence range. Ion modification provided a large improvement in permittivity, especially at low frequencies. The improvement is in a sequence of Cs > Rb ≈ K > Li > Na. In addition to the direct polarity shift caused by the replacement of the OH bond with the O-alkali cation bond, the structure (such as the orientation of the groups) also contributed to the final permittivity. To further understand the relaxation features of the MOFs, a plot of the imaginary part of the complex modulus is presented in Figure 4e. The peak position of this plot implies the relaxation time, and the right shift of this peak indicates a faster polarization process [32]. From this plot we can see that Cs, Rb, and K have the fastest polarization response, followed by Li. However, Na shows less improvement compared with the raw MOF. To summarize, the EIS results indicate that the ion substitution greatly improved the dielectric properties of MOF, including a higher permittivity and relaxation time for a fast polarization response. Among the five alkali elements, K, Rb, and Cs showed the most improvement. Li showed a mediocre performance, and Na showed less change compared with the raw MOF’s dielectric properties.

Based on the above characterizations, one can see that the results are not exactly linear in the sequence from Li to Cs. To figure out the regularities among the phase, structure, and properties, we summarized several data including the XRD data for lattice information, UV–Vis data for energy state, ICP-MS data for the substitution ratio, and the EIS data for dielectric properties. The data were normalized between 0 to 1, as seen in Figure 4f. One can see that the black, grey and yellow lines, corresponding to XRD spacing ratio, EIS permittivity, and EIS relaxion time, respectively, are in a similar trend. This indicates that the ion substitution-induced lattice distortion could be an important factor contributing to the dielectric properties. The XRD spacing ratios calculated from the XRD peak positions indicates a long range-ordered distortion. Such an ordered distortion may lead to a long range ordered polarization, which may be responsible for the changes in permittivity and relaxation time. In addition, for the non-ordered structural defects or deforming, the XRD intensity data (marked in blue) can better reflect the results. One can see that the trend was close to the UV–Vis red curves. This indicates that the new UV–Vis absorption is possibly related to the local bonding features near the alkali ions rather than the long range-ordered lattice distortion.

The final solid-state electrolyte was achieved by ball-milling. The MOFs, magnesium salts, and ether solvent were milled into a homogeneous powder. In this electrolyte, the MOFs severed as a scaffold to improve the chemical stability and to boost the dissociation of the Mg salts; the Mg salts provided the conduction ions, and the ether solvent improved the mobility of the ions. We first optimized the ratio of these three components by measuring the impedance through electrochemical impedance spectroscopy (EIS). As presented in Figure 5a, one can see that the impedance to Mg salt/MOF ratio displayed a volcano shape; the impedance decreased as the Mg salts increased from 1/128 to 1/16, but further increasing Mg salt usage enlarged the impedance. Figure 5b showed the influence of ether usage of impedance. As we expected, more liquid resulted in smaller impedance. The improvement was not linear: the impedance dropped 20-fold from 50 nL mg^−1^ to 125 nL mg^−1^ liquid, but reduced less when a further 150 nL mg^−1^ of liquid was added. Balancing the impedance and reducing the liquid usage, our optimized ratio for electrolyte was 1 salts to 16 weight-ratioed MOF powder, with 125 nL mg^−1^ ether liquid.

We further compared the impedance differences among the different ion substitutions [33,34], as presented in Figure 5c. All electrolytes displayed a semi-circle and a line in the EIS Nyquist plot, and the Bode plots are in Appendix A. The data were fitted in R_1_, R_2_Q, and W equivalent circuits in connected series. R_1_ represents a total ohmic resistance, and R_2_ is the charge-transfer resistance. The total resistance is the sum of R_1_ and R_2_. The R_1_, R_2_, and the calculated conductivity data are presented in Appendix A. The data indicate that the non-modified MOF showed a significantly large R_2_ resistance which was over two orders higher than the other modified MOFs. This implies that the ion substitution strategy can significantly reduce the resistance, especially the charge-transfer resistance of the mixture of electrolytes. One may notice that the R_1_ for raw MOF has a negative value. This is caused by its large R_2_ resistance, which brought a high error margin in both the EIS signal and the following fitting. For the ohmic resistance R_1_, which included the moving resistance for ions and electrons, it gradually decreased from Na to Cs-modified electrolytes, and Li showed a counter-trend lower resistance than Na. The trend of R_1_ inversely followed the interplanar spacing sequence, as confirmed by XRD peak positions in Figure 4f. This indicates that a larger lattice interplanar spacing can benefit the movement of the ion, resulting in a smaller ohmic resistance. For the charge-transfer resistance R_2_, a valley-shaped trend is observed. Among them, the K-modified MOF presented the smallest R_2_ value. The trend of the R_2_ inversely followed the UV–Vis absorption peak position sequence in Figure 4f. In the section above, we concluded that the K-modified MOFs can better stabilize the excited π* state of the ligand frame. We deduced that such an ability can also benefit the stabilization of the charge-transfer intermediate state between Mg and Mg^2+^.

Besides the impedance, the transference number is another critical parameter for ionic conductors. The transference numbers were measured using a potentiostatic polarization method [35,36]. The detailed formula and data are shown in Section 2 and Appendix A. The transference number of Mg^2+^ indicates to which extent Mg^2+^ participates in ion migration. It can also reflect the scaffolds’ ability in benefiting Mg^2+^ HTFI^−^ ion pair dissociation. The results showed an increased transference number following the periodic order, i.e., Li to Cs being 0.047, 0.133, 0.163, 0.220, 0.226, respectively. The non-modified raw MOF showed a value of 0.096. One may notice that the Li modification even showed a transference number lower than the raw MOF. We deduce that in this case, Li^+^ participated in the ion migration, leading to a decreased Mg^2+^ transference number. As we discussed in the above sections, the small size of Li^+^ leads to it being moveably embedded in the lattice, and its decreased repulsion force in distortion leads to a calcination-recovered long range-ordered lattice symmetry. For the other alkali ions, the transference numbers are all improved compared with the raw MOF. Rb and Cs performed the best with the highest transference number, followed by K and Na.

To further explore the relationship between electrochemical performance and other characterization results of bare MOF, a comparation plot was constructed, which can be seen in Figure 5d. In the way that the experiment was designed, the evolution of the lattice structure and the other physical properties showed some similar trends with impedance and the transference number. For example, for the lattice spacing and the ohmic conductance, we can see that the black solid line and the black dotted line showed similar trends. The larger spacing showed a higher ion conductance, which implied that enough interplanar spacing can benefit the ion transfer. The K-modified MOF, which had the largest change in energy state (from the UV data, blue solid line), had the highest charge-transfer conductance (blue dotted line). The positive correlation between the permittivity and the ability for the dissociation of ion pairs was confirmed by the yellow line and yellow dotted line. Li-doping, which involved ion migration, was an exception, as we discussed before. Our findings emphasize the importance of lattice design in ion conductors, which benefited not only the MOF-based solid-state electrolyte but also broadens the field of crystalline electrolytes, including organic polymers and ceramics. We believe that our findings could improve their conduction properties through lattice regulation.

## 4. Conclusions

In this study, we employed an ion substitution strategy to modify the aluminum MOF MIL-121. The aim was to create a scaffold to absorb liquid electrolytes and serve as a quasi-solid-state electrolyte for batteries. The results showed that the alkali ion substitution not only changed the MOFs’ composition but also induced a large-scale change in the lattice, which lead to a profound effect on its permittivity and excited energy state. At 1 Hz, the Cs-modified MOF displayed a permittivity which was over five orders of magnitude higher than the raw MOF. All of the alkali-substituted MOFs showed a red shift in their UV–Vis spectra, caused by their stabilization effect on the excited state. These changes further influenced the electrochemical properties in the final composed electrolyte: the Cs-modified MOF, which had the largest lattice spacing distortion, showed the highest ohmic conductivity of 0.33 mS cm^−1^; the K-modified MOF, which showed the highest red shift in the UV–Vis spectrum, also had the highest charge-transfer conductivity of 0.12 mS cm^−1^; and the highest permittivity from the Cs-modified MOF benefitted the dissociation of ion pairs, which resulted in the highest transference number of 0.226. This work helps to link the relationship between the structure of electrolyte scaffold lattices, physical properties, and electrochemical performance. Long range-ordered structural distortion can benefit ion conduction and enhance permittivity for boosting the dissociation of ion pairs. Defect modification can help stabilize the intermediate state in the electron–ion charge transfer process, and lower the impedance of the entire structure. This work also proved that a low-cost ion substitution effectively changed the electrolyte’s conductivity, which shows its potential for possible scalable industrial applications.

## Data Availability

The data that support the findings of this study are available from the corresponding author upon reasonable request.

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
