# Peer review of "Ion Substitution-Induced Distorted MOF Lattice with Deviated Energy and Dielectric Properties for Quasi-Solid-State Ion Conductor"

_nanomaterials, 2025, doi:10.3390/nano15040274_

Round 1
Reviewer 1 Report
Comments and Suggestions for Authors
The manuscript discusses the lattice change affects MOFs' physical and chemical properties and emphasizes the role of scaffold-salt interaction in the causation of ion-conduction evolution. Results showed that long-range-ordered structure distortion can enhance permittivity for boosting ion-pairs dissociation. The topic is interesting and industrially relevant. Albeit several points explored by the authors are of great interest, some concerns have to be addressed before consideration for publication:
1. Abstract should be improved to highlight the main quantitative achievements of this study.
2. The introduction should emphasize how does this work go beyond the state of the art in this kind of materials.
3. It seems from the XRD that the sample isn’t phase pure as many signals appear. It is advised to add the PDF card number of the reference material and possibly include XRD pattern of the standard in the supporting information.
4. The FTIR spectra assignment can be supported by the following reference doi.org/10.1016/j.heliyon.2024.e40679
5. EIS data discussion should be strengthened by explaining the physical meaning of the equivalent circuit components in the electrochemical system and supported by references such as (Electrochemical, Structural and Thermodynamic Investigations of Methanolic Parsley Extract as a Green Corrosion Inhibitor for C37 Steel) and (Experimental and DFT Atomistic Insights into the Mechanism of Corrosion Protection of Low-Carbon Steel in an Acidic Medium by Polymethoxyflavones from Citrus Peel.
6. The Bode plots can be added in the supporting information.
7. The SI file is missing in the submission
8. Statistical analysis should be considered by reporting standard deviations and error bars if possible.
Reviewer 2 Report
Comments and Suggestions for Authors
The manuscript under review presents an investigation into hybrid solid-state electrolytes containing MOF-based scaffolds and metal salts as ion conductors. The study employs an ion-substitution strategy to manipulate the scaffold structure at the lattice level by replacing hydrogen with larger alkali cations. The review text presented here refers only to the main document, as the SI document was not made available even after I requested it. Below are the detailed points for revision:
1. Language and Sentence Structure: Several parts of the manuscript contain sentences that require revision for clarity and readability. The following lines need particular attention:
- Lines 83-84: The phrasing is ambiguous and should be rewritten for clarity.
- Lines 109-120: The explanation is convoluted and should be restructured to enhance comprehension.
- Line 227: The sentence is confusing and should be rephrased for better readability.
- Line 280 and 283: Requires revision for grammatical correctness and clarity.
- Line 338: The sentence is confusing and should be rephrased for better readability.
2. Figures:
- The figures are too small, making it difficult to discern the presented data. Consider enlarging them for better visualization.
- The SEM image in Figure 2c lacks clarity in the legend and the color scheme. The legend should be revised to improve readability.
- Nyquist plot (Figure 5c): It is recommended to include the equivalent circuit diagram to clarify the presence of R1 and R2 elements.
- The absence of data for all metals in Figure 5c suggests a discontinuity in the frequency range, which should be addressed.
- Most of the raw MOF data is missing from Figure 5c, and its inclusion would provide a complete comparison.
- The distorted semicircles in the Nyquist plot in Figure 5c suggest additional contributions that are not discussed. This should be investigated and explained.
3. Introduction:
- The following reference should be included to provide additional context and relevance to the study: https://doi.org/10.1038/s41467-023-40609-y.
- Consider adding a representation of the MOF MIL-121 structure for a clearer understanding of the scaffold used.
4. Experimental Conditions:
The conditions under which conductivity measurements were performed are not clearly stated. It is important to specify temperature and whether the experiments were conducted in an inert atmosphere.
5. Discussion Section:
- Section 3 lacks a clear and objective discussion that effectively relates all data and explains their true influence on ionic conductivity. A more thorough and structured discussion is necessary to enhance the manuscript’s scientific contribution.
- Were measurements conducted at different temperatures? If so, presenting this data could provide valuable insights.
6. Abstract and Conclusion:
- The last paragraph of the abstract is identical to the conclusion. It is recommended to revise one of these sections to avoid redundancy.
Comments on the Quality of English Language
Grammar should be revised throughout the article. There are sections of the article where the text is quite confusing.
Reviewer 3 Report
Comments and Suggestions for Authors
Paper Title: Ion-Substitution Induced Distorted MOF Lattice with Deviated Energy and Dielectric Properties for Quasi-Solid-State Ion Conductor
Authors: Yike Huang, Yun Zheng, Yan Guo, Qi Zhang, Yingying Shen, Hebin Zhang, Yinan Liu, Yihao Zheng, Pingshan Jia, Rong Chen, Lifen Long, Zhiyuan Zhang, Congcong Zhang, Yuanhang Hou, Kunye Yan, Ziyu Huang, Manting Zhang, Jiangmin Jiang, Shengyang Dong, Wen Lei, and Huaiyu Shao
Affiliations: University of Macau, China University of Mining and Technology, Nanjing University of Information Science and Technology, Wuhan University of Science and Technology
1. Summary of the Paper
The paper investigates the impact of ion substitution on the lattice structure, energy, and dielectric properties of metal-organic frameworks (MOFs) to enhance their performance as quasi-solid-state ion conductors. The authors employ an ion-substitution strategy to manipulate the scaffold structure at the lattice level, leading to changes in dielectric properties and ion conductivity. Through extensive experimental studies, including XRD, FTIR, SEM, and electrochemical impedance spectroscopy, the paper successfully demonstrates the potential of ion-substituted MOFs to achieve improved electrochemical performance. The research findings provide valuable insights into scaffold-salt interactions and their role in advancing solid-state electrolyte technologies.
2. Review Criteria
2.1 Clarity of Aim
-
Evaluation: The aim of the study is clearly stated, focusing on the impact of ion substitution on MOF properties to enhance ion conduction.
-
Recommendation: While the aim is well-articulated, a brief discussion on the broader impact of these findings in the context of energy storage systems would add value.
2.2 Content Structure
-
Evaluation: The paper is well-structured, progressing logically from background and methodology to results and conclusions.
-
Recommendation: Improving transitions between sections, particularly from synthesis to characterization, would enhance readability.
2.3 Methodology
-
Evaluation: The methodology is robust, detailing synthesis procedures, characterization techniques, and electrochemical testing protocols comprehensively.
-
Recommendation: Justification of experimental conditions, such as temperature and ion concentration choices, would strengthen the reproducibility of the study.
2.4 Verification and Case Studies
-
Evaluation: Experimental results validate the proposed hypotheses, with comparisons to existing MOF-based electrolytes providing credibility.
-
Recommendation: Including additional tests under varied environmental conditions (e.g., humidity, temperature extremes) would enhance applicability.
2.5 Graphs and Diagrams
-
Evaluation: Figures and tables are clear and effectively convey key findings. XRD and SEM images provide crucial structural insights.
-
Recommendation: Adding schematic diagrams to visualize ion substitution mechanisms would further improve reader comprehension.
2.6 Results and Discussions
-
Evaluation: The results are well-presented, with thorough discussions linking experimental findings to theoretical expectations.
-
Recommendation: Expanding on the practical implications of these findings in battery technology would add depth to the discussion.
2.7 Conclusions
-
Evaluation: The conclusions effectively summarize the study's findings and suggest future directions.
-
Recommendation: More specific future work suggestions, such as scalability challenges or potential industrial applications, would be beneficial.
2.8 Overall Quality
-
Evaluation: The paper is of high quality and provides a significant contribution to the field of solid-state electrolytes.
-
Recommendation: Minor grammatical errors and typographical issues should be corrected to enhance readability.
3. Recommendations for Publication
Based on the evaluation, I recommend the following actions:
-
Minor Revisions Required
-
Provide additional justification for experimental assumptions and conditions.
-
Expand the discussion on real-world applications and potential scalability.
-
Include additional graphical representations to improve visualization.
-
Perform thorough proofreading to correct minor language errors.
-
Comments on the Quality of English Language
Minor grammatical errors and awkward phrasing should be corrected
